# In-Situ Oxidative Polymerization of Pyrrole Composited with Cellulose Nanocrystal by Reactive Ink-Jet Printing on Fiber Substrates

**DOI:** 10.3390/polym14194231

**Published:** 2022-10-09

**Authors:** Xu Li, Meijuan Cao, Shasha Li, Luhai Li, Yintang Yang, Ruping Liu, Zhicheng Sun, Lixin Mo, Zhiqing Xin, Yinjie Chen, Yaling Li, Yi Fang, Yuansheng Qi

**Affiliations:** 1Beijing Engineering Research Center of Printed Electronics, Beijing Institute of Graphic Communication, Beijing 102600, China; 2JiangXi LianSheng Electronic Co., Ltd., Jingdezhen 333002, China

**Keywords:** PPy, CNC, fiber substrate, conductivity, ink-jet printing

## Abstract

A simple and novel method for the deposition of polypyrrole (PPy) and cellulose nanocrystal (CNC) composites on different fiber substrates by reactive ink-jet printing was proposed. PPy/CNCs composites were successfully prepared, and the surface resistance of conductive layer deposited on different fiber substrates is the least when the monomer concentration is 0.6 M. PPy/CNCs were deposited on polyethylene terephthalate (PET) to form a conductive layer by adding polyvinyl alcohol (PVA), and the optimum sintering temperature is 100 °C (monomer/PVA ratio 4.0, conductivity 0.769 S cm^−1^). The PPy/CNCs conductive layer deposited on the paper has the lowest surface resistance and the best adhesion, and the surface resistance of PPy/CNCs conductive layer decreases first and then increases with the increase of sulfonate concentration. Moreover, the volume of anion in sulfonate will affect the arrangement and aggregation of PPy molecular chain in composite materials. Appropriate sulfonate doping can improve the conductivity and stability of conductive paper, and the maximum conductivity is 0.813 S cm^−1^. Three devices based on PPy/CNCs conductive paper were proposed and fabricated. Therefore, this ink-jet printing provides a new method for the preparation of conductive materials, sensors, energy storage and electromagnetic shielding, etc.

## 1. Introduction

Conductive polymer, also known as conjugated polymer, is a kind of material whose conductivity is between semiconductor and conductor after doping. Since Shirakawa et al. [1] discovered polyacetylene as a conductive polymer in 1977, the long-held view that polymer materials are insulators has been broken. Subsequently, PPy [2], polyaniline [3], polythiophene [4] and other types of conductive polymers are constantly developed. These conductive polymers not only have the characteristics of polymer materials, but own the conductivity of semiconductors and metals, which have been widely used in the fields of electrostatic shielding, sensing materials and energy storage materials [5,6,7].

PPy is one of the most promising conductive polymers, which has the advantages of simple preparation, low cost, good thermal stability, low density, high dielectric constant and high conductivity [8]. However, pure PPy is rigid, difficult to dissolve in common organic solvents, and has poor mechanical ductility, lacking more prominent electrical, optical and biological properties, making it difficult to process into desired products. Many researchers have found that the composite of PPy with other functional materials (carbon materials, metals, inorganic materials and organic materials) can make up for the shortcomings and deficiencies of single materials, and greatly broaden the application scope of PPy. For example, Li et al. [9] prepared polyaniline/PPy composite nanofibers with core-shell structure by covering the surface of polyaniline with PPy thin layer, which can be used as electrode materials for high-performance supercapacitors in neutral aqueous solution. Lin et al. [10] developed a Raman photoacoustic active probe based on PPy-polydopamine complex. Wang et al. [11] used PPy and Rhodamine B as raw materials to construct biocompatible bifunctional nanoparticles by one-step polymerization. Among the many materials, CNC is the most common organic material composite with PPy [12,13,14]. PPy/CNCs composites not only possess the environmental stability and conductivity of PPy, but also own the thermal stability, mechanical strength and flexibility of fiber materials [15,16]. Hanif et al. [12] developed a light-weight aerogel compression strain sensor based on PPy/CNCs composite material. The sensor exhibited good performance under repeated compression cycles and resistivity changes under different loads.

One of the most common methods for preparing conductive layer of fiber substrates is to directly coat conductive materials on fiber substrates. This method usually results in low electrical conductivity and poor mechanical properties, which is attributed to the shedding of the conductive layer from the substrate or poor adhesion. An interesting alternative is to prepare the conductive polymers by in-situ polymerization of monomers on the fibers directly and combined with ink-jet printing, which can not only control the thickness and shape of the conductive layer, but also avoid the contact between the deposition system and the substrate and reduce waste. This technology is promising for the production of ink-jet printable inks such as PPy dispersion for polymer sensor devices [17,18] or PPy nanofiber crystal composite dispersion [19]. Hohnholz et al. [20] developed a very simple method to obtain custom patterns from conductive polymers using a standard laser printer. Weng et al. [21] detailed the most common printing methods used to make conductive polymers, including PPy, such as screen printing, rotary printing, and ink-jet printing. Stempien et al. [22] prepared a PPy fabric coating for electromagnetic shielding by ink-jet printing. At present, ink-jet printing technology is considered as one of the most industrially promising conductive layer manufacturing methods, which is widely used in fiber, textile and other flexible electronics.

In this work, we present a simple and low-cost method for producing PPy/CNCs conductive layer using dual-nozzle ink-jet printing. In the deposition process of PPy, the reaction takes place on the surface of CNC, and the hydrogen bonds between N atoms in pyrrole (Py) and H atoms in the fiber will result in the CNC being wrapped by PPy and deposited on the fiber substrate. Among the five fiber substrates, PPy/CNCs deposited on the paper substrate has the least surface resistance and the best adhesion. Three sulfonates are selected as the dopant, and the surface resistance of PPy/CNCs paper base conductive layer is measured to see whether it can be used to optimize the conductivity of PPy/CNCs. On the one hand, appropriate sulfonate doping can improve the conductivity and stability of the conductive layer. On the other hand, too high sulfonate concentration or too large anion volume will negatively affect PPy/CNCs conductivity. Moreover, PPy/CNCs were deposited on a PET substrate by adding PVA, and the surface resistance of the conductive layer is minimized at 100 °C. Three devices based on PPy/CNCs conductive paper were prepared.

## 2. Materials and Methods

### 2.1. Reagents and Material

Chemicals including Py, ammonium persulfate (APS) and CNC (200 nm in length, 10 nm in outside diameter) were supplied by Macklin, (Shanghai, China). Specifically, Py was freshly distilled for all the following experiments. PVA, *p*-Toluenesulfonic acid sodium salt (*p*-TSA), sodium dodecyl benzene sulfonate (SDBS) and sodium polystyrene sulfonate (PSS) were bought from Aladdin (Shanghai, China). In addition, PPy was synthesized through an oxidation reaction between Py and APS. The aqueous solutions of Py (1.342 g, 0.2 M) were cooled under an ice bath. Then the solution of APS as an oxidation (4.564 g, 0.2 M) was slowly added into the reaction solution. Subsequently, the resulting reaction mixture was stirred for another 2 h. The resulting black PPy precipitate was filtered, and washed with ethanol (Jindongtianzheng Precision Chemical Reagent Factory, Tianjin, China), followed by water, and finally with acetone (Beijing Chemical Works, Beijing, China). Finally, PPy powder was dried in air and then vacuum dried at 60 °C over 12 h, which is stored and directly used in the following experiments.

### 2.2. Deposition of PPy/CNCs Conductive Layer on the Substrate

The formation of PPy conductive layer was achieved by the oxidative ink-jet printing of Py with APS as an oxidant [23]. The reaction equation is shown in Figure 1. The setup of ink-jet printing technique for the PPy/CNCs conductive layer is summarized in Figure 2. Typically, the PPy/CNCs conductive layer can form on the surface of the substrate or permeate the internal structure via the reactive ink-jet printing technique. In this work, a prototype of the digital ink-jet printer (ReaJet SK 1/080, Frankfurt, Germany) was employed. The in-kjet printing system is consisted of two nozzle systems, and each nozzle system is equipped with an ink tank to store the reactants. The ink pool of nozzle 1 was filled with the ink composition prepared by the mixed aqueous solution of Py and CNC, and the ink pool of nozzle 2 was filled with the aqueous solution of APS. Specifically, the molar ratio for the Py and oxidant is determined as 1:1 during the printing process. First, the mixture of Py and CNC was released by nozzle 1 onto the surface of the fiber substrate and seeps into the interior of the fiber substrate. Next, the oxidant APS was released to the surface and internal structure of the fiber substrate by nozzle 2 following the path of nozzle 1. Simultaneously, in situ polymerization occurs on the surface and internal structure of the fiber substrate, and the deposited conductive layer can form a strong mixed structure with the fiber substrate, which greatly increases the adhesion and conductivity of the conductive layer. In the deposition of non-fiber substrate (PET), the first nozzle was loaded with Py and CNC mixed aqueous solution. However, the second nozzle was loaded with APS and PVA mixed aqueous solution. Furthermore, the substrate should be placed on a thermostatic heater for the printing operation. The PPy/CNCs conductive layer was robustly deposited on PET, which is attributed to the bonding effect of PVA.

### 2.3. Characteristics

Polarizing optical microscope (LEICA DM2700M, Wetzlar, Germany) and scanning electron microscopy (SEM, SU8020, Hitachi, Japan) were employed to determine the morphology PPy/CNCs and the distribution of PPy on the parent material surface. PPy/CNCs powder samples were diluted with pure water, treated with sonication, then dropped onto mica sheets and dried at low temperature. Next, the particle size and morphology of PPy/CNCs composites were observed by atomic force microscopy (AFM, Park NX-wafer, Seoul, Korea), and the morphology roughness of samples was analyzed by XEI software (Park Systems Crop, 5.1.4.build3, Korea). Fourier transform infrared spectroscopy (FTIR) spectra of PPy/CNCs composites were recorded by an Nicolet 380 FTIR spectrometer (Madison, WI, USA) using the standard KBr disc method at a resolution of 4 cm^−1^ over a wavelength region from 4000 to 400 cm^−1^. The Raman spectra of PPy/CNCs were collected through the Renishaw inVia (Renishaw, London, UK), and the excitation wavelength of Raman spectra was 785 nm. The acquired Raman spectral range was set from 400 to 2000 cm^−1^ at a resolution of 2 cm^−1^. In addition, organic element analysis (Vario EL cube, Elementar, Langenselbold, Germany) was used to determine the weight ratios of C, H, O, N, and S elements in PPy/CNCs composites. For specific operation, PPy/CNCs samples were first burned or decomposed into gas at high temperature, analyzed by a purge-capture adsorption desorption method on three special columns, and then detected by thermal conductivity detector. The surface resistance of PPy/CNCs deposited on each fiber substrate and PET conductive layer was measured by RTS-9 four-probe tester (Guangzhou Four Probe Electronic Technology Co., LTD, Guangzhou, China), and the conductivity is calculated by measuring the thickness, and surface resistance. The adhesion of the conductive layer on each fiber substrate was analyzed by comparing the ratio between the weight of the conductive layer after multiple washing and the initial deposition weight. The influence of different sulfonates on the surface resistance of paper substrate was also studied, and the I-V curve and stability of electrical conductivity was measured.

## 3. Results and Discussion

### 3.1. Structure and Wrapping Structure of PPy/CNCs

CNC is a kind of nanoscale cellulose extracted from natural fibers. The length of CNC is about 200 nm and the diameter is 10 nm. N element in PPy can easily form hydrogen bonds with H element in CNC. A strong intermolecular force makes PPy adhere to the surface of CNC and to form a wrapping structure [24]. In dual-nozzle ink-jet printing, the nozzle has an aperture of 50 microns, which is about 250 times the length of CNC. Therefore, we expected that the printing work smoothly. Studies have reported that CNC has a Chiral nematic liquid crystal phase structure in aqueous solution, and different surface charge types of materials affect the self-assembly performance of CNC. The assembly modes of CNC composite material are shown in Figure 3e,f, which are n-type and p-type, respectively. The negatively charged material particles are parallel to the long axis of CNC and do not affect the chiral nematic ordering of CNC. Therefore, in n-type co-assembly method, CNC molecules are orderly arranged in layer upon layer to form composite materials. In contrast, the addition of positively charged material particles destroy the chiral nematic structure of CNC due to electrostatic interaction and particle flocculation. Therefore, p-type co-assembled CNC molecules are distributed disordered and superimposed layer by layer to form composite materials [25]. In this work, the co-assembly of CNC composites is different from the previous n-type or p-type co-assembly. Instead, a strong intermolecular hydrogen bonding force is used to co-assemble the composite materials of PPy and CNC. In the presence of the oxidant APS, Py is polymerized on the CNC surface to form a powerful PPy package, which is attributed to the effect of hydrogen bonding (see Figure 3g). At the same time, PPy in PPy/CNCs composites also forms strong hydrogen bonds with the fiber substrate, which makes PPy/CNCs firmly attached to the fiber substrate [26]. The PPy/CNCs will readily be stacked layer upon layer to form composite materials conductive layer through order arrangement, disorder arrangement or orderly disorder coexistence (see Figure 3h,i).

### 3.2. Polarizing Optical Microscope Images of PPy/CNCs Deposited the Substrate

Initially, the optical images of various materials deposited with PPy/CNCs were obtained. As shown in Figure 4, the PPy/CNCs printed on different fiber substrates demonstrated the great uniformity. With the same concentration of Py, a large number of conductive layer of PPy/CNCs were deposited on paper, non-dust cloth, linen, cotton and wool substrates. PPy/CNCs are easily deposited and strongly adsorbed on the fiber substrate due to the abundant hydrogen bonds. Particularly, PPy/CNCs on paper and non-dust cloth show excellent air permeability and adsorption effect (see Figure 4a,b) [27]. PPy/CNCs can also readily deposited on linen, cotton and wool, but the adsorption is poor as compared to that on paper and non-dust cloth (see Figure 4c–e). After washing, most of the sediment was washed away and only part of the PPy/CNCs fabric structure formed on the surface of linen, cotton and wool was retained [28,29]. Paper is the internal fibers intertwined together, Py adsorption on the fiber, deposition into PPy/CNCs, will form a solid fiber and PPy/CNCs mixed structure. Non-dust cloth is a kind of craft woven fiber fabric, but the neat arrangement of the fibers and the treatment of the fiber surface will lead to the adhesion of the deposited PPy fiber composite structure is worse than that of the paper. Linen, wool and cotton are natural fiber structures, which are easy to deposit a large amount of PPy/CNCs. However, linen, wool and cotton fibers are loosely stacked and wound without a strong winding braided structure. Therefore, the PPy/CNCs conductive layer cannot be firmly adsorbed.

### 3.3. Surface Morphology

The SEM images of the PPy/CNCs composites were further examined for the samples prepared from Py and CNC ink oxidized by APS. In Figure 5a, a solid wrap composite was obtained through the surface wrapping of PPy on CNC, in which the resulting composited materials were stacked and intertwined with each other to form a whole. With the amplified region (see Figure 5b), the wrapping structure of PPy and CNC seemed to be crosslinked and piled up to form a large number of holes, which can readily absorb electromagnetic induction and electromagnetic waves as electromagnetic shielding materials. When the image size was enlarged to 10,000× and 20,000× (see Figure 5c,d), the wrapping structures were clearly seen to be cross-linked and stacked together such as small tentacles. Therefore, these SEM results provide direct evidence that PPy was successfully wrapped on the surface of CNC to form a wrap. Specifically, PPy has excellent electrical conductivity, CNC has a great adaptability to the composite material. The combination of PPy and CNC is expected to maintain a certain adhesive strength, to improve the performance of the composite material. Accordingly, through the combination with the properties of PPy and CNC, the mixed material with the wrap structure will be useful for the applications including sensing [30], energy storage [31], electrostatic shielding [32], electrode materials [33,34] and conductive materials [35].

The AFM technology was used to observe the particle size and morphology of CNC, PPy and PPy/CNCs composites at the molecular level. The AFM image of CNC is shown in the Figure 6a. The length of CNC is about 200 nm, and the outer diameter is about 20 nm. The R_a_ value is 1.13 nm, and the maximum height R_p_ value is 21.1 nm, which is consistent with the theoretical value of CNC [36]. PPy particles were formed by Py polymerization, and the R_a_ value is 0.329 nm (see Figure 6c). The surface roughness of PPy particles is low, which is due to the small gap between the long chains of molecules [37]. An image of the PPy/CNCs is shown in the Figure 6e. Theoretically, PPy/CNCs composites are formed by stacking PPy wrapped on the surface of CNC, and the surface roughness should be between CNC and PPy. The R_a_ value of PPy/CNCs is 0.631 nm, which is larger than PPy but smaller than CNC, and the actual value is consistent with the theory. In addition, the R_p_ value of PPy/CNCs is 35.53 nm, which is larger than the sum of the R_p_ values of CNC and PPy. This should be the formation of hydrogen bonds between PPy and CNC molecules, which increased the molecular spacing, resulting in a larger R_p_ value of PPy/CNCs.

### 3.4. Organic Element Analysis

After the double-nozzle printing, the PPy/CNCs composites were cleaned with deionized water, ethanol and acetone for several times to remove the impurities. Then the resulting sample was left in drying oven at 50 °C over 24 h, to completely remove the solvents. The organic element analysis was carried out to determine the elements distribution. The corresponding weight percentages of C, H, O, N and S elements are shown in Figure 7a. Theoretically, PPy/CNCs composites are composed with PPy and CNC. In the process of polymerization, two α-protons close to the N atoms on the Py ring are removed due to the conjugation of oxidation reaction. In addition, the obtained PPy molecules will be accompanied by sulfuric acid complexes and impurities. The oxidation produces sulfuric acid and ammonium sulfate are considered as by-products, which are expected to be removed by repeated washing with various solvents.

The corresponding results of the elemental analyses of the chemically oxidative polymerization of PPy and PPy/CNCs are summarized in Table 1. The elemental analysis contained S element, which is attributed to the synthesis of PPy doped with sulfate. The N/S molar ratio significantly indicates the doping level for one pyrrole ring in the PPy chain. It can be assumed that this fabricated PPy has a doping level of eight pyrrole rings for one dopant, as calculated from experimental data [38]. In addition, the N/S molar ratio of PPy/CNCs is consistent with that of PPy by experimental data calculation. Due to CNC does not contain N and S elements, which has no effect on the N/S ratio of composite materials. However, the C/N and C/S molar ratios obtained in PPy/CNCs are significantly different from those obtained in PPy, which may be caused by the C element in CNC in the composite material. These results show that CNC and PPy are encapsulated together to form PPy/CNCs composites [15,16].

In Figure 7b, relatively, the mole of N element is 1.0474, while N element only exists in PPy. In PPy, the mole of C element is 4 times of N element, so the mole of C element in PPy is 4.1896. PPy and CNC both contain the C element, and the mole of C element is 4.4835. Therefore, the mole of C element in CNC is 0.2939. Both PPy and CNC are high molecular weight polymers. In this work, the molecular weights of PPy and CNC are considered to be equal. Under this premise, the weight of PPy is 21.3828 times than that of CNC by calculation. During the preparation, 2.0 g Py and 0.10 g CNC were added. In fact, the weight of Py is 20 times than that of CNC. According to the above discussion, theoretical calculation results were in a good agreement with those of experimental data.

### 3.5. FTIR Spectra

The FTIR spectra of the CNC, PPy and PPy/CNCs are shown in Figure 8. As for the CNC powder, the O-H tensile vibration peak at 3424 cm^−1^, the C-H symmetric tensile vibration peak at 2899 cm^−1^, and the shoulder peak at 1164 and 1116 cm^−1^ are caused by C-O antisymmetric vibration stretch and C-OH skeleton vibration, respectively. At 1062 and 1033 cm^−1^, there are obvious pyranose ring skeleton vibration peaks, which are the characteristic peaks of cellulose. Glycoside-ch deformations with ring vibrations and -OH bending are shown at 896 cm^−^^1^, which are characteristic of β-glycosidic bonds connecting glucose molecules in CNC structure [39,40]. As for the PPy powder, the N-H stretch band of PPy is observed at around 3438 cm^−1^, and the weak bands at 2926 and 2852 cm^−1^ are related to the stretching vibration of C-H bond. The peak at 1637 cm^−1^ represents the stretching of C=N bond. The characteristic peaks at 1538 and 1453 cm^−1^ correspond to the C=C stretching and N-C stretching modes of the PPy ring, respectively [41]. In addition, peaks at 1301 and 1042 cm^−1^ are ascribed to in-plane C-H deformation and in-plane N-H stretching vibrations. Particularly, these strong peaks at 1168 and 891 cm^−1^ demonstrate the doping state of PPy [42].

In the FTIR spectra of PPy/CNCs composites (see Figure 8), the presence of strong characteristic bands of PPy at 1538, 1453, 1168 and 1042 cm^−^^1^ provide the direct evidence for the successful formation of PPy/CNCs composites. In contrast to the spectra of PPy, the wave peaks of PPy/CNCs move toward large wavelengths as a whole, which is the result of the composite of PPy and CNC. There is a wide N-H peak at 3420 cm^−^^1^, which should be caused by the combination of PPy and CNC. In the PPy/CNCs composites, some bands in FTIR spectra are close to the parent components, indicating that the structure of PPy/CNCs is close to that of the PPy itself [43,44].

### 3.6. Raman Spectra

The Raman spectra of the CNC, PPy and PPy/CNCs are shown in Figure 9. As for the CNC powder, the shear vibration of C-H at 1480 cm^−^^1^, the bending vibration peak of C-H at 1381 and 1340 cm^−^^1^, the C=C asymmetric stretching vibration peak at 1150 cm^−^^1^, and the in-plane sway vibration peak of C-H at 995 and 974 cm^−^^1^. The symmetric stretching vibration of glycosidic bond C-O-C at 1121 cm^−^^1^, the asymmetric stretching vibration peak of glycosidic bond C-O-C at 1097 cm^−^^1^ [45,46]. As for the PPy powder, the peak at 1584 cm^−^^1^ belongs to the C=C stretching vibration peak of the oxidation state component, the peak at 1506 cm^−^^1^ belongs to the C=C stretching vibration peak of the neutral state component, and the deformation vibration peak in C-H plane at 1252 cm^−^^1^. The two peaks at 1050 and 1078 cm^−^^1^ belong to the in-plane deformation vibration of C-H, and the other two peaks at 1384 and 1330 cm^−^^1^ belong to the stretching vibration of the ring. The ring deformation vibration of PPy at 927 cm^−^^1^, and the stretching vibration peak of C-N at 1434 cm^−^^1^ [47].

Furthermore, the Raman spectra of PPy/CNCs composites was measured (see Figure 9). The C=C stretching peak at 1598 cm^−^^1^ is considered to be an overlap of the two oxidized structures, which should be attributed to the conversion of C=C from the neutral state component to the oxidized state component after the combination of PPy and CNC. The peak at 1381 and 1316 cm^−^^1^ can be assigned to antisymmetric C-H in-plane bending, and the deformation vibration peak in C-H plane at 1243 cm^−^^1^ [45,47]. The ring deformation vibration of PPy at 932 cm^−^^1^, and the stretching vibration peak of C-N at 1474 cm^−^^1^. The two peaks at 1078 and 1049 cm^−^^1^ belong to the in-plane deformation vibration of C-H. In contrast, some bands in the Raman spectra of PPy/CNCs composites are close to the spectra of PPy, indicating that the structure of PPy/CNCs is close to the structure of PPy itself.

### 3.7. Surface Resistance

#### 3.7.1. The Surface Resistance of PPy/CNCs Deposited Different Fiber Substrates

PPy/CNCs deposited on different fiber substrates played a significant role in the conductivity of the fiber substrates. Therefore, it will be greatly important to explore the surface resistance on different fiber substrates varies with the concentration of Py. Specifically, PPy/CNCs composites were deposited on five fiber substrates with the ink-jet method via adjusting the concentration of Py. The corresponding surface resistance with different substrates and varied concentration was measured (see Figure 10a). Clearly, the surface resistance of PPy/CNCs composites deposited on each fiber substrate decreased with the increase of the concentration of Py. When the monomer concentration is 0.6 M, the surface resistance of conductive layer deposited on different fiber substrates is the least. The surface resistance of conductive paper substrate is less than that of other fiber substrates, due to the interwoven and intertwined fibers in the paper. In addition, Figure 10b shows the ratio of weight after PPy/CNCs deposition (multiple washes) to initial deposition weight under different fiber substrates and various Py concentrations. The weight ratio can indicate the strength of adhesion of PPy/CNCs deposited on different fiber substrates. Most of the PPy/CNCs deposited in the dust-free cloth and linen were removed after several washing due to the limitation of the material properties. Moreover, the vast majority of PPy/CNCs deposited on cotton and wool were removed after many washes, leaving only the surface attached parts. In contrast, the fibers in the paper are interwoven and intertwined, resulting in a lot of pore structure, which is conducive to the firm binding of PPy/CNC on the paper substrate, forming a solid paper hybrid structure [48,49]. This paper hybrid structure not only obtains high electrical conductivity but possesses great adhesion of conductive layer to paper substrate. Therefore, the PPy/CNCs conductive layer deposited on the paper base is difficult to clean off, which makes its weight ratio larger than several fiber substrates.

#### 3.7.2. Surface Resistance of PPy/CNCs Deposited PET Substrate

Furthermore, we explored the conductivity of PPy/CNCs that varied with the Py/PVA weight ratio on PET substrate. In Table 2, different Py/PVA weight ratios were adjusted to explore the influence of Py/PVA weight ratio on the conductivity and adhesion of PPy/CNCs deposited on PET substrate. From Formula 1 to Formula 5, the conductivity of PPy/CNCs increases with the weight of Py, and the maximum conductivity can reach 0.769 S cm^−1^. The adhesion on PET film is enhanced with the increase of PVA content. In this work, we optimized the different weights of Py, APS and PVA to seek good conductivity and adhesion. In Formula 3, PPy/CNCs deposited films show great conductivity and excellent adhesion on PET substrate, and the conductivity is 0.625 S cm^−1^.

Temperature sintering destroys the coating structure of resin and conductive paste, and makes the connection distance and connection mode between conductive paste change, which will significantly affect the spreading film formation and electrical conductivity of conductive layer. The effect of temperature on deposition of PPy/CNCs on PET substrate was also investigated. In Figure 11a, PPy/CNCs were deposited on PET substrate, and the temperature of thermostatic heater was studied from 30 to 150 °C. Four-probe tester was used to determine the surface resistance of film after sintering at different temperatures, to fully explore the sintering temperature with the minimum surface resistance. Specifically, the surface resistance decreases with the increasing of temperature from 30 to 100 °C. The minimum surface resistance is obtained at 100 °C. From 100 to 150 °C, the surface resistance increased with the increasing of temperature. In the low temperature sintering stage, PVA possibly wrap the conductive paste, blocking the connection between the conductive paste. As the temperature rose, the conductive paste is coalesced and connected with together. The barrier effect of PVA is weakened, causing the reduction of the surface resistance value. With a high temperature (>100 °C), the conductive paste is too converged, resulting in the formation of gap between the conductive paste [50,51]. Typically, the gap becomes larger with the increase of temperature, so the surface resistance will increase.

In Figure 11b, the sintering time of PPy/CNCs deposited films is observed to decrease with the increase of temperature. In five formulations, with the same sintering temperature, the more content of PVA added in different formulations, the longer the sintering time would be. At the low temperature stage, the sintering time between the two temperature gradients varied greatly. In addition, the sintering time is also affected by temperature and PVA content. In the high temperature stage, the sintering time is mainly influenced by temperature, and the content of PVA in different formulations show little influence.

#### 3.7.3. Effect of Different Sulfonates on Surface Resistance of PPy/CNCs Deposited on Paper Substrate

The concentration of sulfonate affects the ability of conjugated structures in PPy to transfer electrons, and the anions in sulfonates affects the arrangement and aggregation of PPy molecular chains as counter ions in the composite materials [52,53,54]. Therefore, the type of sulfonate and wt% can significantly affect the conductivity of PPy. To study the influence of sulfonate on the surface resistance of PPy deposited on the paper substrate, different types and various wt% sulfonates were added to ink pool of nozzle 1 for the fabrication of PPy.

As shown in Figure 12, for all three sulfonates, the surface resistance of PPy/CNCs decreases first and then increases with the increase of sulfonate concentration. Differently, the minimum surface resistance of 221 Ω/sq is obtained with the addition of about 2 wt% *p*-TSA, and the conductivity is 0.452 S cm^−1^ by calculation (see Table 3). When the addition of SDBS is 1 wt%, the minimum surface resistance is 203 Ω/sq, and the conductivity is 0.492 S cm^−1^ by calculation (see Table 3). The minimum surface resistance is 123 Ω/sq with the addition of 1 wt% of PSS, and the calculated conductivity is 0.813 S cm^−1^ (see Table 3). In this work, the surface resistance first decreased and then increased with the increase of sulfonate concentration, which is attributed to the fact that the introduction of sulfonate could effectively enhance the electron transfer ability of PPy conjugated π-bond and improve the electrical conductivity. If the concentration of sulfonates is too high, it may cause the conjugated π-bond to be less able to transfer electrons and reduce the electrical conductivity.

In fact, different sulfonates will also significantly affect the electrical conductivity of PPy. The small volume of anions in sulfonates is conducive to the mutual aggregation of PPy molecular chains into a more perfect conductive structure, which is also shown as the gradual increase of electrical conductivity on the macro level. The larger doped anion volume leads to the increase of the separation degree between PPy molecules and the interchain jump of carriers, so the electrical conductivity decreases. In contrast, the anion volume in PSS and SDBS is larger than that in *p*-TSA (2 wt% sulfonate), so the conductivity of *p*-TSA doped PPy is larger than that of other two sulfonates, which is manifested in the small surface resistance of *p*-TSA doped.

The I-V curve test of PPy/CNCs conductive paper samples with or without doping can prove sulfonate dopant can improve the conductivity and conductive stability of conductive paper. In the I–V curves of Figure 13a and Figure 14a, both the undoped and the sulfonate-doped samples all show an ohmic behavior. With the same voltage, the doped current is almost an order of magnitude larger than the undoped one, indicating that the doped sulfonate can improve the conductivity. In Figure 13b and Figure 14b, the conductivity of the undoped sulfonate decreases significantly after 1 day while the conductivity of the doped sulfonate decreases significantly after 10 day. Moreover, the electrical conductivity changes of doped and undoped sulfonates are relatively stable, and there is no order of magnitude change. The conductivity is observed to slowly decrease, which may be a natural phenomenon.

#### 3.7.4. Paper-Based Device

In this work, the fabricated conductive paper was used to fabricate three paper-based devices (see Figure 15). In Figure 15a,b, the fabricated paper is demonstrated to be conductive when a LED light connected with a voltage supplier. Figure 15c,d show that the fabricated conductive paper may be employed as an accelerometer. Specifically, when a force is applied on the end of the paper sheet to bend the paper device with a certain angle, a resistance change is observed during the bending due to piezo-resistivity [48]. Particularly, the resistance change is directly proportional to the bending angle (see Figure 15d). In addition, in Figure 15e,f, we have also demonstrated that the acid pH sensor can be made by monitoring the resistance value of the paper base deposited by PPy/CNCs. When the acid toxic gas volatilizes, the resistance value will change to play an early warning role [55].

## 4. Conclusions

In summary, the novel PPy/CNCs composites were successfully developed through the chemical oxidation of Py and CNC by APS via reactive ink-jet printing at room temperature. The optimal conditions for deposition of PPy (oxidizer/monomer ratio 1.0, monomer concentration 0.6 M) on various fiber substrates were obtained for reactive ink-jet printing, in which PPy/CNCs deposited on paper substrates obtained the least surface resistance and the best adhesion. On the PET substrate, PPy/CNCs conductive layer was successfully deposited by adding PVA, and the optimal sintering temperature is 100 °C (Py/PVA ratio 4.0, conductivity 0.769 S cm^−1^). On the paper base, appropriate sulfonate doping can improve the conductivity and stability of the conductive layer, and the maximum conductivity is 0.813 S cm^−1^ with the addition of 1 wt% of PSS. However, too high sulfonate concentration or too large anion volume of sulfonate will negatively affect PPy/CNCs conductivity. Three PPy/CNC composites show great applications in paper devices. This reactive ink-jet printing provides a new method for sensor preparation, energy storage and electromagnetic shielding.

## Figures and Tables

**Figure 1 polymers-14-04231-f001:**
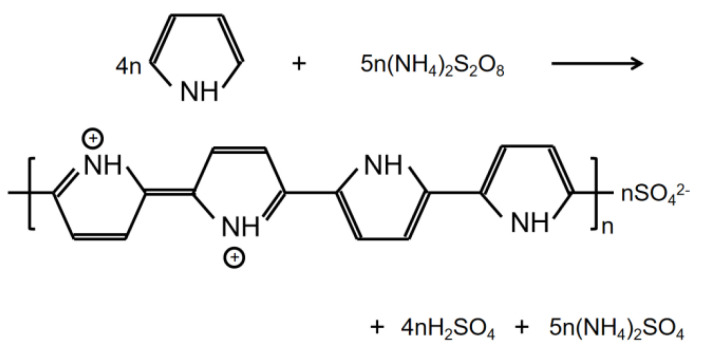
The possible reaction mechanism of Py and APS.

**Figure 2 polymers-14-04231-f002:**
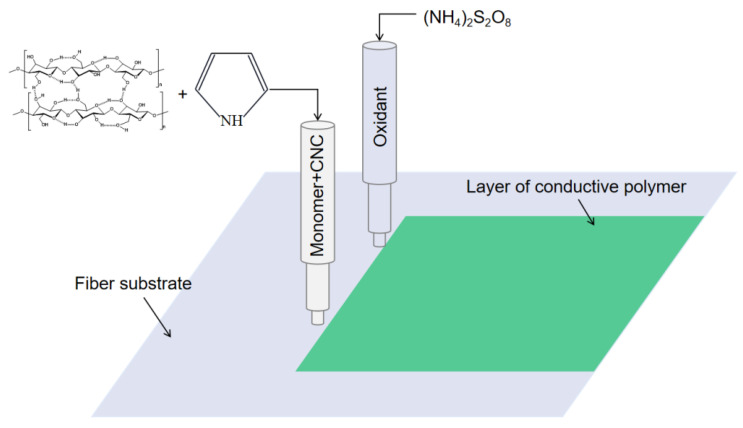
The deposition of PPy/CNCs on fiber substrate by ink-jet printing.

**Figure 3 polymers-14-04231-f003:**
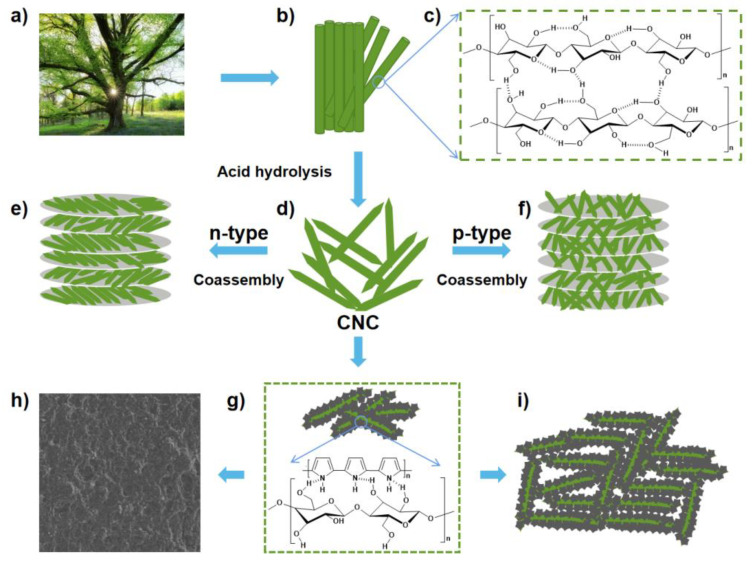
Structure and wrapping structure of PPy/CNCs. (**a**) Tree; (**b**) Cellulose fiber; (**c**) The molecular structure of cellulose; (**d**) CNC obtained by acid hydrolysis. (**e**) The co-assembly process of CNC and n-type material. N-type material refers to the material mainly conductive by electrons. CNC molecules are orderly arranged in layer upon layer in composite materials. (**f**) The co-assembly process of CNC and p-type material. P-type material refers to the material mainly conductive by holes. CNC molecules are distributed disordered and superimposed layer by layer to form composite materials. (**g**) The wrapping structure of PPy/CNCs and the possible bonding modes of hydrogen bonds. (**h**) PPy/CNCs composite conductive layer; (**i**) Possible sedimentary structure of PPy/CNCs.

**Figure 4 polymers-14-04231-f004:**
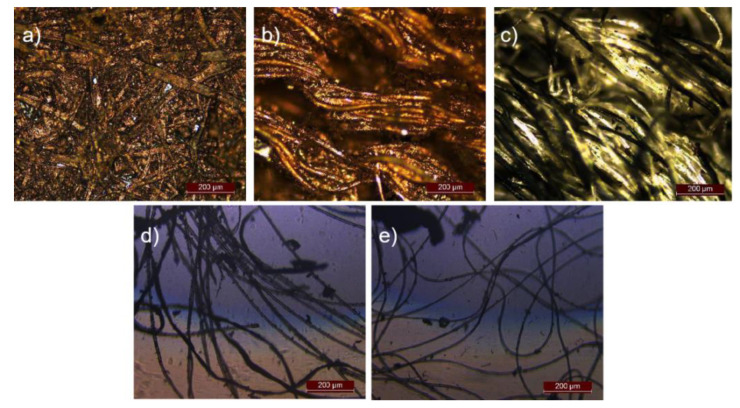
Polarizing optical microscope images of PPy composite (**a**) paper; (**b**) non-dust cloth; (**c**) linen; (**d**) cotton; (**e**) wool.

**Figure 5 polymers-14-04231-f005:**
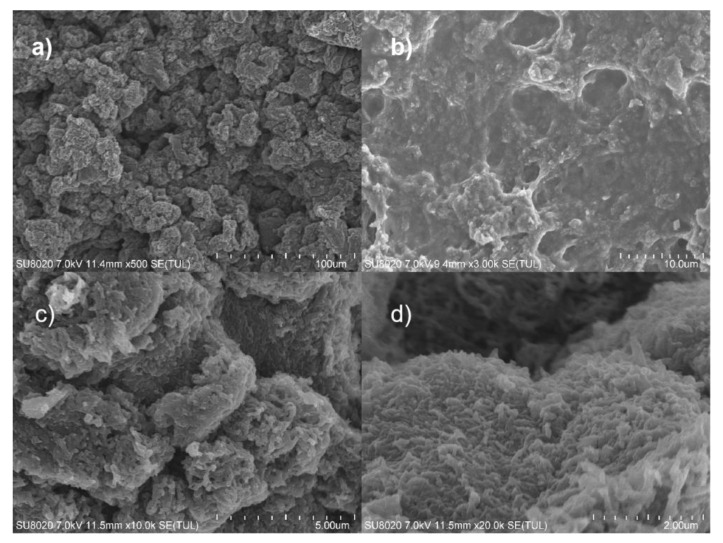
SEM images of PPy/CNCs (**a**) 500×; (**b**) 3000×; (**c**) 10,000×; (**d**) 20,000×.

**Figure 6 polymers-14-04231-f006:**
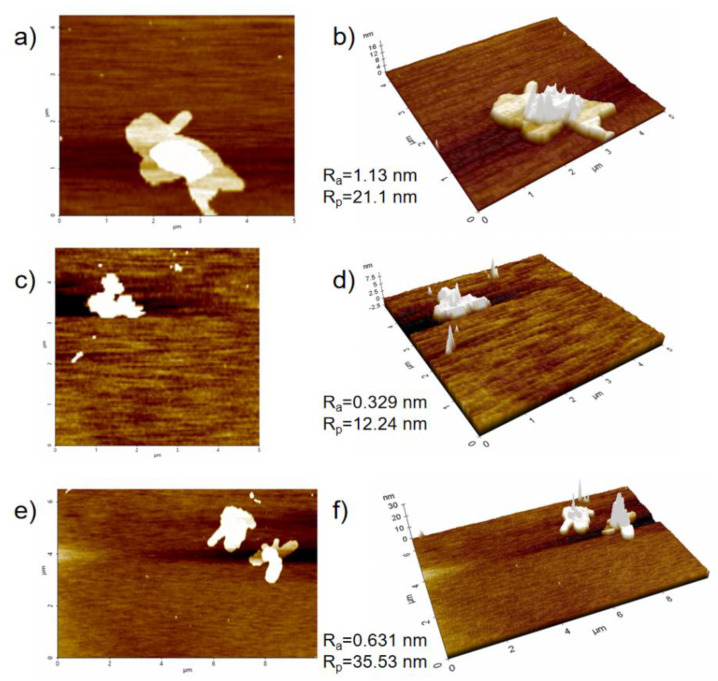
AFM images of PPy/CNCs (**a**) CNC; (**b**) Three dimensions of CNC; (**c**) PPy; (**d**) Three dimensions of PPy; (**e**) PPy/CNCs; (**f**) Three dimensions of PPy/CNCs.

**Figure 7 polymers-14-04231-f007:**
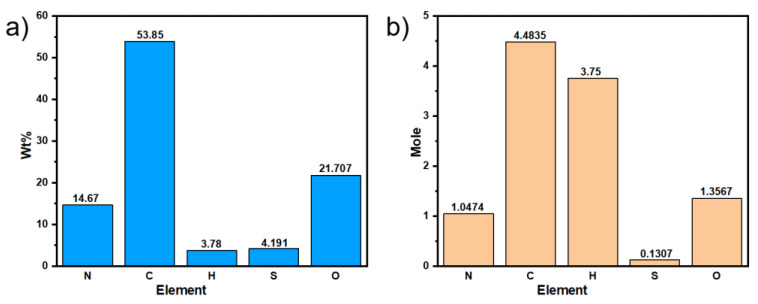
Each element of PPy/CNCs (**a**) wt% (The content of each element is less than 1% error); (**b**) mole.

**Figure 8 polymers-14-04231-f008:**
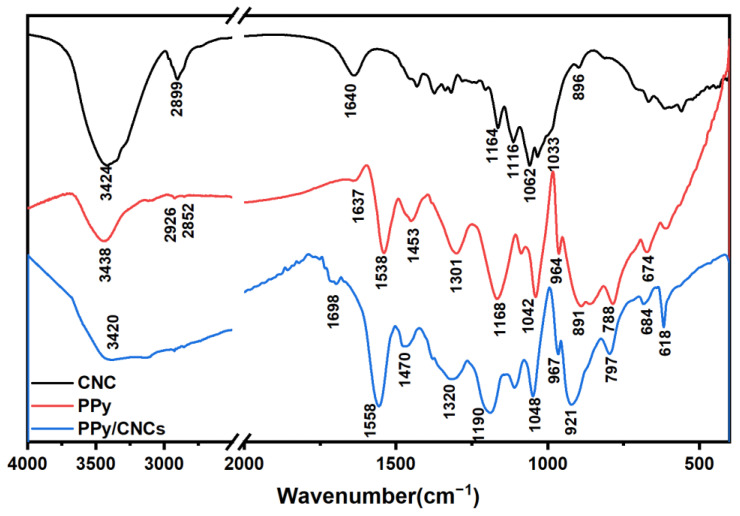
FTIR spectra of CNC, PPy and PPy/CNCs.

**Figure 9 polymers-14-04231-f009:**
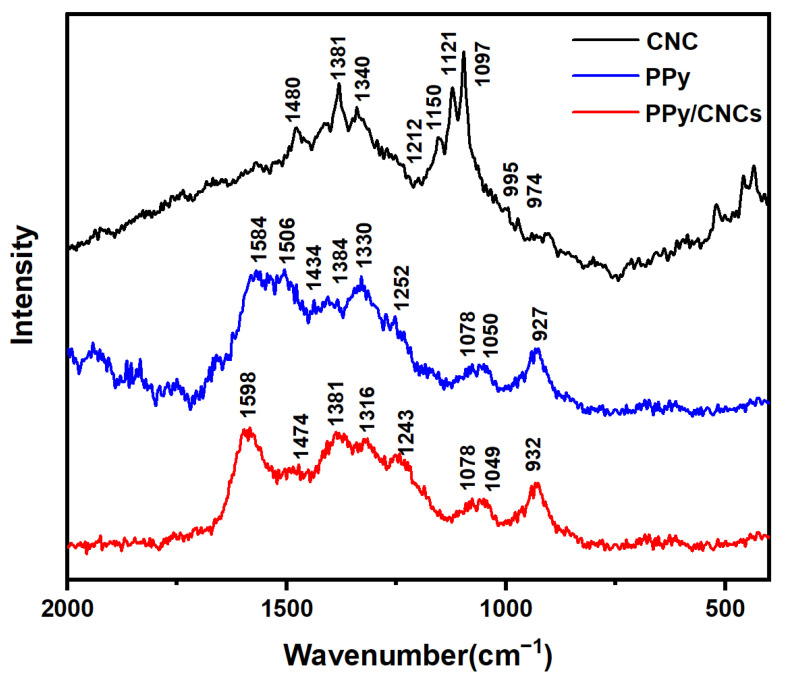
Raman spectra of CNC, PPy and PPy/CNCs.

**Figure 10 polymers-14-04231-f010:**
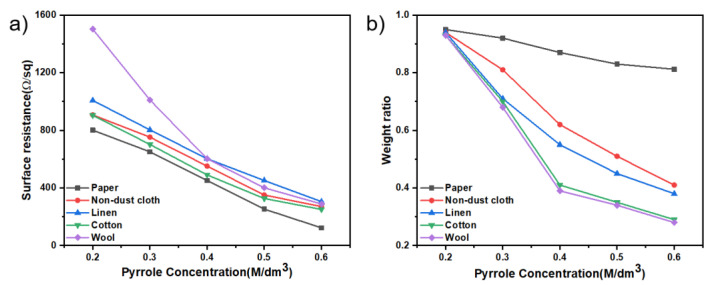
PPy/CNCs deposited different fiber substrates (**a**) Surface resistance on different fiber substrates varies with the concentration of Py. (**b**) Ratio between the weight of the conductive layer after multiple washes and the initial deposited weight, on different fiber substrates and different Py concentrations.

**Figure 11 polymers-14-04231-f011:**
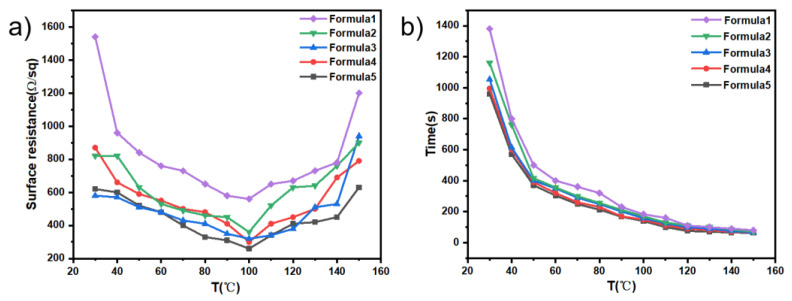
Effect of temperature on deposition of PPy/CNCs on PET substrate (**a**) surface resistance; (**b**) sintering time.

**Figure 12 polymers-14-04231-f012:**
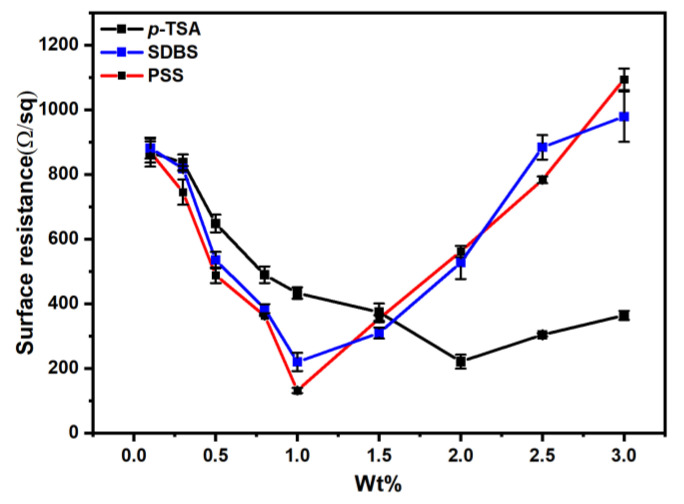
Effect of type and wt% of sulfonate on the surface resistance of deposited PPy/CNCs.

**Figure 13 polymers-14-04231-f013:**
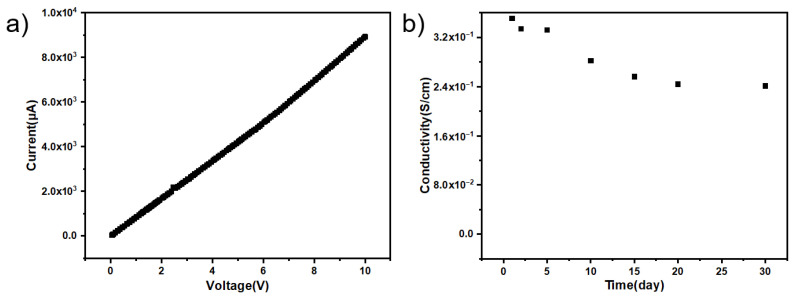
In-situ polymerization deposition of PPy/CNCs on paper substrate (**a**) I−V curve of a conductive paper (30 mm long, 5 mm wide and 100 μm thick). (**b**) Conductivity stability of the conductive paper.

**Figure 14 polymers-14-04231-f014:**
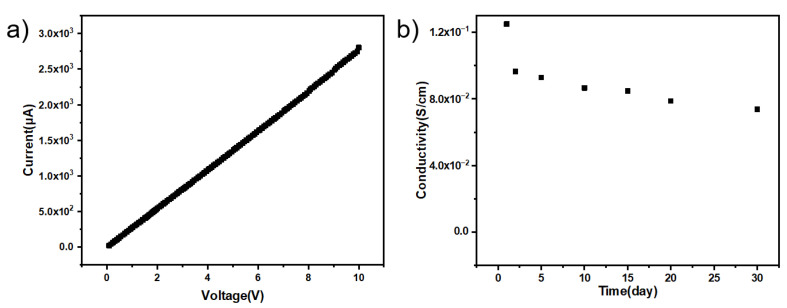
In-situ polymerization deposition of PPy/CNCs doped sulfonate on paper substrate (**a**) I−V curve of *p*-TSA-doped conductive paper (30 mm long, 5 mm wide and 100 μm thick). (**b**) Conductivity stability of the conductive paper.

**Figure 15 polymers-14-04231-f015:**
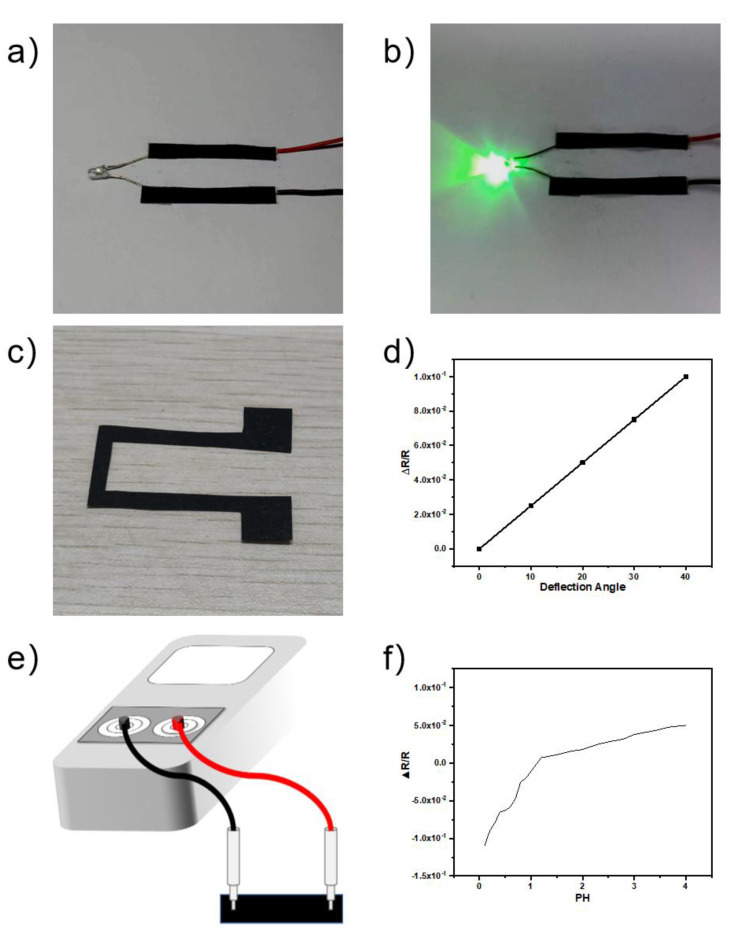
Three paper-based devices (**a**,**b**) Conductive paper as electrodes in a LED light device. (**c**,**d**) A simple accelerometer design made of the conductive paper, and the resistance change upon bending of the paper device. (**e**,**f**) Monitoring the resistance value of the paper base deposited, and the resistance change upon pH of the paper device.

**Table 1 polymers-14-04231-t001:** Elemental analysis (wt%, molar ratio) of PPy and PPy/CNCs.

Sample	C	N	S	C/N	C/S	N/S
**PPy**	50.92	15.55	4.43	3.82	30.65	8.02
**PPy/CNCs**	53.85	14.67	4.191	4.28	34.30	8.01

**Table 2 polymers-14-04231-t002:** Conductivity of PPy/CNCs varies with the Py/PVA weight ratio.

Sample	Py (g)	APS (g)	PVA (g)	Weight Ratio of Py/PVA	Conductivity (S cm^−1^)
**Formula1**	1.2	4.08	1.5	0.8:1	0.357
**Formula2**	1.6	5.44	1	1.6:1	0.556
**Formula3**	1.8	6.12	0.75	2.4:1	0.625
**Formula4**	1.92	6.53	0.6	3.2:1	0.667
**Formula5**	2	6.8	0.5	4:1	0.769

**Table 3 polymers-14-04231-t003:** Effect of different sulfonates on the conductivity of PPy/CNCs.

Sulfonate	Chemical Formula	Optimum Doping Concentration	Conductivity (S cm^−1^)
***p*-TSA**	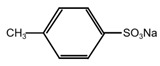	2%	0.452
**SDBS**	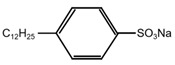	1%	0.492
**PSS**	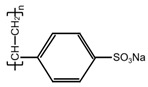	1%	0.813

## Data Availability

Not applicable.

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
