# Peer review of "In-Situ Oxidative Polymerization of Pyrrole Composited with Cellulose Nanocrystal by Reactive Ink-Jet Printing on Fiber Substrates"

_polymers, 2022, doi:10.3390/polym14194231_

Round 1
Reviewer 1 Report
The manuscript discusses an interesting approach for deposition of PPy on various substrates by using double nozzle printing. Still, the manuscript needs to be corrected, before that it can not be accepted.
-English style and grammar must be extensively corrected.
-Introduction should be rewritten in a clear way, and redundancy should be reduced throughout the manuscript.
-Abstract, correct "ammonium disulfite", and the sentence in lines 24-28 should be rewritten in a clear style.
-Specify the abbreviation mentioned for the first time throughout the paper.
- The proposed polymerization reaction equation (Fig. 1) shows that the stoichiometric ratio of oxidant/pyrrole is 1.25. Please explain why the mole ratio of 1 was used.
-In the characterization section, the Authors should specify all the methods used and add the instruments’ specifications.
-Please explain what you mean by the doping methods of cellulose (Line 215).
-The two sentences in lines 246-250 are somehow contradicting each other, explain.
-There is a problem with data collection of FTIR analysis, some characteristic peaks have disappeared, for instance, the N-H stretching band of PPy around 3400 cm-1, the same in the case of the composite, peaks around 3700 look like noises with low intensities.
-The interpretation of FTIR must be corrected, and the band position values in the text must be the same on the spectra. Polymer, 2019, 174, 11-17.
-Please specify the excitation wavelength used for collecting Raman spectra.
-It will be better to use either resistivity or conductivity for all the results.
Reviewer 2 Report
The research presented in this work have a significant application. I suggest some corrections of the text, before further procedure.
Please see the list of remarks and suggestions that would help to improve the text.

Reviewer 3 Report
I have attached a PDF document.

Round 2
Reviewer 1 Report
The Authors have improved the manuscript, however, still there are some comments must be addressed:
- What do you mean by conducting polymers are constantly new (Line 33)?
- Line 40, “lacking in electrical, optical and biological properties” is that true?
- Line 57-59, rewrite.
- Still it is not clear why the Authors used an oxidant/pyrrole mole ratio of 1, while the authors themselves proposed a polymerization reaction equation (Fig. 1) that shows that the stoichiometric ratio of oxidant/pyrrole is 1.25 to prepare doped PPy.
- Line 257, persulfate S2O82- should be sulfate group, also “Specifically, one anion (S2O82-) of APS can coordinate with the positive charge of six Py rings”, is not matching with the proposed chemical reaction in Figure 1.
- Still the Authors did not specify all the methods used and the specifications of the instrument, please specify for every single method used, for instance, AFM analysis, which equipment, model, substrate, sample preparation, software for roughness calculation, and so on.
- Table 1, please add the units for each column, wt%, mole% and check the ratio calculations of elemental analysis.
- Fig. 3g, correct pyrrole structure to polypyrrole according to caption and text.
- Correct the unit of the excitation wavelength of Raman spectra.
- Still the co-assembly mechanism of CNC composites with different surface charge fillers should be rewritten in clear language.
Reviewer 3 Report
I am pretty happy to see all the changes. The authors need to find a better way to address Question 10. Question 10:Page 16, Line 552: Please provide any theoretical explanations for the linear relationship. It can't explain how the resistance changed linearly with the bending angle.
According to the authors, whether it is n-doping or p-doping depends on the co-assembled materials. This seems irrelevant to the paper. Why do the authors used a whole paragraph talking about n/p-doping. Explain
Round 3
Reviewer 1 Report
The manuscript can be accepted in the present form.
Author Response
Thank you very much for your suggestions on this manuscript.